# In Vitro Antibacterial Activity of Ethanolic Extracts Obtained from Plants Grown in Tolima, Colombia, Against Bacteria Associated with Bovine Mastitis

**DOI:** 10.3390/vetsci12090903

**Published:** 2025-09-18

**Authors:** Yeimy Lorena Robledo-Díaz, Aurora Alejandra Sánchez-Varón, Yeli Camila Van-arcken Aguilar, María del Pilar Sánchez-Bonilla, Jorge Enrique Hernández-Carvajal

**Affiliations:** 1Medicina Veterinaria y Zootecnia, Facultad de Medicina Veterinaria y Zootécnia, Universidad Cooperativa de Colombia, Ibagué 730006, Colombia; lorenarobledo582@gmail.com (Y.L.R.-D.); alejanchez30@gmail.com (A.A.S.-V.); yeli.vanarcken@campusucc.edu.co (Y.C.V.-a.A.); maria.sanchez@campusucc.edu.co (M.d.P.S.-B.); 2Química Farmacéutica, Facultad de Ciencias Básicas, Universidad Santiago de Cali Sede-Pampalinda, Calle 5 # 62-00, Cali 760036, Colombia; 3Tecnología en Regencia de Farmacia, Instituto de Educación a Distancia, Universidad del Tolima-Sede Cali, Carrera 5-N #61N-126 Barrio Flora Industrial, Cali 760032, Colombia

**Keywords:** bovine mastitis, antimicrobial activity, medicinal plants, toxicity, natural products

## Abstract

Mastitis is a bacterial disease that affects dairy cattle, causing high costs. Antibiotics used in pharmacological treatments usually have limited efficacy, and their indiscriminate use can lead to the presence of residues in milk and its derivatives, which represents a potential risk to public health by favoring the development of bacterial resistance. For this reason, alternative treatments are required, one of them being active pharmaceutical ingredients obtained from medicinal plants that require validation of their antibacterial activity. This research presents information on the antibacterial activity of ethanolic extracts of five medicinal plants cultivated in Tolima, Colombia, against bacteria associated with mastitis. The results showed a high antibacterial potential for the extracts of *Psidium guajava* L., *Rosmarinus officinalis* L., and *Calendula officinalis* L. against the mastitis pathogens evaluated.

## 1. Introduction

Globally, there is a high demand for milk production due to its nutritional relevance in the human diet. However, several conditions negatively affect both the quantity and quality of the product, with mastitis being one of the main causes. This disease represents one of the most frequent and costly pathologies in the dairy industry, with economic losses estimated at approximately USD 22 billion per year globally [1]. It is estimated that about 70% of economic losses in dairy production systems are associated with mastitis, due to significant reductions in production, which can range from 233 to 1799 L per affected cow [2]. Additionally, there are increased costs associated with clinical veterinary follow-up, the use of pharmacological treatments, decreased milk quality, increased risk of animal slaughter, and high replacement costs [2].

Bovine mastitis is an inflammatory pathology of high incidence in dairy cattle. This disease is associated with the most common etiological agents (e.g., microorganisms such as *S. aureus*, *Mycoplasma bovis*, *Corynebacterium bovis,* and *S. agalactiae*), while *Escherichia coli*, *Streptococcus dysgalactiae*, *Streptococcus uberis,* and *Klebsiella* spp., especially *K. pneumoniae*, are examples of environmental pathogenic bacteria [3,4]. Some microorganisms, such as *S. aureus* or *S. agalactiae,* have adapted to survive within the mammary gland and can be transmitted from one cow to another, mainly during milking [5,6]. Other microorganisms, such as *Streptococcus uberis*, *E. coli*, and other coliforms, come from the contaminated environment and are opportunistic invaders during milking, especially in the dry period [3]. All of these microorganisms in clinical mastitis significantly affect milk composition [7,8,9].

In Colombia, the prevalence of subclinical mastitis at the herd level was 50% and showed a direct association with somatic cell count (SCC) and bacterial count [10]. Factors associated with the presence of subclinical mastitis included multiparity, prolonged lactation, lack of pre-sealing, and overmilking [10].

The microorganisms responsible for generating bovine mastitis have developed resistance to traditional treatments due to the transmission of resistant genes, which have been reported in new generations of microorganisms associated with mastitis [11]. In the cases of *S. aureus* and coagulase-positive *Staphylococcus*, bacteria resistant to multiple drugs were isolated in milk samples because they express surface proteins associated with adhesion to surfaces, promotion of biofilm formation, invasion of epithelial cells, or immune evasion. An example is the resistance to penicillin, with percentages of 25.5% and 29.1%, respectively, which can be associated with the production of β-lactamases, which are responsible for hydrolyzing β-lactam antibiotics such as penicillin [3,12].

The *S. uberis* strain showed resistance to erythromycin (24%) and tetracycline (37.5%), while *S. dysgalactiae* showed 43.2% resistance to tetracycline [12]. On the other hand, resistance in *E. coli* to ampicillin (24%) and tetracycline (23.6%) was recorded [12]. As for *K. pneumoniae*, 100% resistance to beta-lactam antibiotics was observed, as well as to chloramphenicol (57%), streptomycin (43%), and sulfamethoxazole–trimethoprim (43%). This behavior has been related to the presence of fimbrial genes (fimA, mrkA, and mrkD), as well as the ecpA gene, which are associated with both multidrug resistance and biofilm production in clinical bovine mastitis [3,7,9,11,12].

In the treatment of mastitis, it is permissible to use drugs such as pirlimycin, cloxacillin, novobiocin, amoxicillin, dihydrostreptomycin, penicillin G, erythromycin, and cefapirin. However, resistance to these drugs, adverse reactions in animals, and antibiotic residues in milk and its derivatives are concerns. Alternatives include vaccination, nanotechnology, probiotics, stem cell therapy, antimicrobial peptides, photodynamic therapy, antibody treatments, and natural products (NPs) [13,14].

The NPs, when used in combination with drugs, offer advantages because they reduce the duration of treatment and are less toxic; this could be an option for the treatment of bovine mastitis. An advantage of NPs is that they are multicomponent mixtures that can act synergistically by employing different mechanisms of action on microbial cells or indirectly by interfering with cell adhesion to surfaces and/or bacterial cell communication [13,14].

Therefore, there is a need to look for new therapeutic alternatives, among which the use of natural products of vegetable origin as a potential source of active pharmaceutical ingredients against pathogenic microorganisms stands out [15,16]. Colombia has a wide plant biodiversity, estimated at approximately 50,000 species, many of which produce secondary metabolites with diverse bioactive properties of pharmaceutical interest [17].

In this sense, it has been reported that the ethanolic extract of *Calendula officinalis* presented bactericidal activity against *Staphylococcus aureus* MSSA 25923L (IZ = 28 ± 2 mm) as well as other microorganisms such as *S. aequatoria*, *E. coli*, *C. tropicalis*, and *C. famata*. This activity has been attributed to the presence of phenolic carboxylic acids and flavonoids [18,19]. Similarly, ethanolic extracts of *P. guajava* L. have shown activity against methicillin-resistant *Staphylococcus aureus* (IZ = 25 mm) and *E. coli* (IZ = 15 mm), an activity related to the presence of flavonoids such as reynoutrin and tannins [20,21]. In methanolic extracts of *Matricaria chamomilla* L., phenolic compounds, such as protocatechuic acid and quercetin, have been identified, which are associated with low antibacterial activity against *Staphylococcus aureus* and moderate activity against *E. coli* [22]. Likewise, the ethanolic extract of *Rosmarinus officinalis* L. flowers has shown activity against *S. aureus* (IZ = 7.5 ± 0.7 mm), attributed to the presence of phenolic compounds such as carnosic acid, carnosol, and rosmarinic acid [23]. Finally, in aqueous or methanolic extracts of *Carica papaya* L., the flavonoids quercetin and kaempferol have been identified, which may be associated with antibacterial activity, reporting minimum inhibitory concentrations against *E. coli* (10 mg/mL) and *S. aureus* (2.5 mg/mL) [24].

The objective of this research was to generate information on the antibacterial activity of the ethanolic extracts obtained from plants cultivated in Tolima that have demonstrated a history of antibacterial activity against coagulase-positive *Staphylococcus* and *Streptococcus* spp. strains associated with bovine mastitis.

## 2. Materials and Methods

### 2.1. Collection of Plant Material

The plants were selected from crops in the Department of Tolima, which had a history of antibacterial activity and the presence of phenolic compounds [19,21,25,26,27,28,29]. The material was collected during the rainy season. The corresponding specimens were deposited in the Toli Herbarium of the Universidad del Tolima (Ibagué-Colombia), where taxonomic determination was performed (Table 1).

### 2.2. Preparation of Ethanolic Extracts

The plant material was subjected to a rigorous cleaning process to ensure its purity and to eliminate any potential phytosanitary concerns. Thereafter, the material was subjected to a drying process that utilized a circulating air oven at a temperature range of 45–50 °C for a period of 48 h. Subsequently, the particle size was reduced using a semi-industrial blender. The extraction of the five selected species was achieved through the utilization of 96% ethanol as a solvent, employing the maceration method [30]. The ground plant material of each sample (150 g) was then exposed to 450 mL of 96% ethanol (1:3) at room temperature for a period of 10 min at hourly intervals. Following a period of one hour, the ethanolic extract was removed by decantation, and the solvent (96% ethanol) was replenished. The extract obtained was stored in an amber flask with a capacity of 1 L. This process was repeated on nine occasions until the exhaustive extraction of secondary metabolites was confirmed by thin-layer chromatography (TLC).

The exhaustive extraction of the samples was monitored by detecting steroid compounds with a mobile phase of ethyl acetate:hexane (2:8) [31,32]; the derivatizing solution was vanillin in orthophosphoric acid. The final extracts were subjected to a process of evaporation, whereby they were reduced to a dry state. Final extracts were evaporated to dryness using a Buchi R-114 rotary water bath, and finally in a vacuum oven at 40 °C to constant weight.

### 2.3. Detection of Flavonoids Using Thin Layer Chromatography (TLC)

The presence of flavonoids was determined by thin-layer chromatography (TLC). Silica gel 60 F254 plates were utilized as the stationary phase, while the mobile phase was composed of a mixture of ethyl acetate, acetic acid, formic acid, and water, in proportions of 100:8:4:8. Rutin was utilized as the reference substance. The derivatizing agent employed was diphenyl boric acid aminoethyl ester (NP developer) in conjunction with polyethylene glycol 4000 (PEG 4000). The plates developed with the derivatizing substance were observed under ultraviolet light (365 nm). These conditions were utilized to identify characteristic stains to flavonoids using the developer diphenyl boric acid aminoethyl ester [31,33].

### 2.4. In Vitro Antimicrobial Activity

#### 2.4.1. Biological Samples

Based on the research conducted by Sánchez et al. [34], the biological material used to isolate the Streptococcus and *Staphylococcus aureus* microorganisms for the biological assays in this study was collected. The procedure is described below. The biological material used for the microbiological tests was obtained from 100 cows belonging to a dairy herd in the municipality of Anaime, Tolima, Colombia [34]. It should be noted that this material was from a previous study and was not part of the original results reported in this work. The California test was used to diagnose bovine mastitis [35,36], after which positive milk samples were extracted for bacterial isolation. These samples were refrigerated at 2–4 °C and transferred, while maintaining the cold chain, to the multifunctional laboratory at the Universidad Cooperativa de Colombia in Ibagué for processing. The samples were cultured and the bacterial microorganisms identified, yielding *Streptococcus* spp. and coagulase-positive *Staphylococcus* as working strains.

The samples were inoculated using the surface seeding technique and a 0.01 mL calibrated loop on blood agar and MacConkey agar media. The samples were then incubated at 37 °C for 24–48 h under aerobic conditions. The isolates obtained were characterized morphologically by studying the colonies and performing Gram staining, following the methodology described by Markey et al. [37]. Gram-positive cocci were subjected to the catalase test to differentiate between the *Streptococcus* and *Staphylococcus* genera. The tube coagulase test was then used to identify coagulase-positive staphylococci (CPS), including *S. aureus*, and differentiate them from coagulase-negative staphylococci (CNS) [34].

It is important to mention that molecular identification of the microorganisms was not performed because they were phenotypically grouped as coagulase-positive *Staphylococcus*. The Gram-positive cocci that presented characteristic morphology of dewdrop growth on blood agar and were negative to the catalase test were identified as *Streptococcus* spp. The Gram-positive cocci, grouped in clusters, isolated from colonies that grew on blood agar but not on MacConkey agar, showing golden coloration, beta-hemolysis, and positive to the catalase test, were associated with the genus *Staphylococcus*. Isolates identified as coagulase-positive were selected for further evaluation against the extracts [34].

The samples were collected during the morning milking routine, and no experimental procedures were conducted. Consequently, no ethics committee approval was sought. Informed consent was obtained for the collection of samples [34].

#### 2.4.2. Evaluation of Antimicrobial Activity by Disc Diffusion

The evaluation of the bactericidal activity of the ethanolic extracts was carried out using the Kirby-Bauer disk agar diffusion method in Mueller–Hinton medium, following the guidelines established by the Clinical and Laboratory Standards Institute (CLSI) [38]. This methodology facilitates the evaluation of the in vitro susceptibility of microorganisms to pure substances or mixtures of compounds, including plant extracts and semi-purified fractions [39]. Bacterial suspensions were adjusted to the 0.5 McFarland scale, corresponding to an approximate concentration of 1.5 × 10^8^ CFU/mL [40]. The inoculation process entailed the utilization of a sterile swab that had been impregnated with each bacterial suspension. This was then employed to achieve even distribution of the bacterial suspensions over the agar surface, with the mass sweep seeding technique being used to facilitate this. Subsequently, six sterile filter paper disks were distributed evenly across the agar medium. Each disk was impregnated with 30 μL of the ethanolic extracts of the different samples (30 mg of dry extract dissolved in 1 mL of 10% dimethyl sulfoxide). The plates were then subjected to an incubation process at a temperature of 37 °C for a duration of 24 h. At the conclusion of the incubation period, the diameter of the inhibition zones was measured and expressed in millimeters (mm). For each bacterial strain, three independent experiments were conducted, each comprising five replicates. An erythromycin sensi-disc (10 mg/mL) was utilized as a positive control.

#### 2.4.3. Statistical Analysis

The data on the antibacterial activity of the ethanolic extracts are presented as the mean of three independent experiments, each with five replicates, with the standard deviation. A completely randomized design was employed, alongside one-way analysis of variance (ANOVA) at a 95% confidence level and a significance level of α = 0.05 (5%). Statistical tests were conducted to verify the normality of the data. The null hypothesis (H0) was rejected if *p* < 0.05. The descriptors were identified using the following letters: A = 0–9 mm, B = 10–20 mm, C = 21–30 mm, and D = >30 mm. Each letter indicates a range that expresses the independent variable inhibition halo in millimeters (mm). The free statistical software InfoStat (version 2020) was used [41].

##### Ethical Aspects

Informed consent was obtained from the proprietors of the farm. Samples were collected during the morning milking without any experimental procedures; therefore, the approval of an ethics committee was not requested. The sampling and laboratory techniques were performed in accordance with the parameters established by the National Mastitis Council (NMC) [42].

### 2.5. Brine Shrimp Lethality Bioassay

The lethality bioassay of *Artemia salina*, a simple and inexpensive model that is readily available, was performed. The toxicity test was adapted to laboratory conditions in accordance with the protocol established by Mclaughlin et al. [43]. *Artemia salina* larvae were obtained in a 3.8% sea salt solution, and 48 and 72 h old nauplii were used in the bioassays after the commencement of incubation. In order to undertake the toxicity evaluation of the ethanolic extracts, a stock solution was prepared. This was achieved by taking 20 mg of the plant extract, 0.5 mL of DMSO, and 1.5 mL of distilled water (for a total of 2 mL of solution). Subsequently, dilutions of 1000 µg/mL, 100 µg/mL, and 10 µg/mL were obtained from this solution. A control was prepared by the addition of 0.5 mL of DMSO and 4.5 mL of seawater (3.8%) to achieve a total volume of 5 mL. In the biological test for each vial, 10 nauplii were utilized (30 nauplii per dilution). The count of dead organisms was enumerated 24 h after seeding.

Preliminary information regarding the safety of the primary extracts of the medicinal plants evaluated was sought using the biological model. A proportional analysis was performed, with the concentration (µg/mL) designated as the independent variable and the percentage of mortality as the dependent variable.

## 3. Results

### 3.1. Phytochemical Evaluation

The percentage yields of the ethanolic extracts of the plant species used were evaluated (Appendix A). The preliminary phytochemical evaluation by thin-layer chromatography (TLC) was undertaken with the objective of detecting the presence of phenolic compounds, including flavonoids. The R_f_ values and coloration of the spots are reported in Table 2 and Figure 1; the ethanolic extracts analyzed exhibited a mixture of compounds of varying polarities, including flavonoids. This assertion is substantiated by the observation of orange spots under ultraviolet light (365 nm) subsequent to development with the NP-PEG reagent [31,33].

### 3.2. In Vitro Antibacterial Activity of the Ethanolic Extracts Against Coagulase-Positive Staphylococcus and Streptococcus spp.

The results of the in vitro antibacterial activity of the ethanolic extracts are shown in Table 3 and Appendix A. The study revealed that the activity of secondary metabolites on coagulase-positive *Staphylococcus* and *Streptococcus* spp. strains resulted in zones of inhibition measuring (13 ± 2.2–21 ± 3.2 mm) and (14 ± 3.2–21 ± 1.9 mm), respectively. The investigation established that ethanolic extracts of *P. guajava* L. (21 ± 3.2) and *R. officinalis* L. (19 ± 2.1) leaves exhibited the most effective activity against coagulase-positive *Staphylococcus*. The ethanolic extracts of the flowers of *C. officinalis* L. (21 ± 1.9), leaves of *R. officinalis* L. (17 ± 2.9), and *P. guajava* L. (15 ± 2.3) were found to be the most active against *Streptococcus* spp. In contrast, the ethanolic extract of the aerial parts of *M. chamomilla* L. did not demonstrate activity against the evaluated strains. Furthermore, the ethanolic extract of the leaves of *C. papaya* L. demonstrated no activity against coagulase-positive *Staphylococcus*. The erythromycin-positive control exhibited a zone of inhibition measuring 32 ± 4.5 mm and 31 ± 4.5 mm for the coagulase-positive *Staphylococcus* and *Streptococcus* spp. strains, respectively. The results demonstrate that the ethanolic extracts of the flowers of *C. officinalis* L. and the leaves of *P. guajava* L. contain active substances against *Streptococcus* spp. and coagulase-positive *Staphylococcus*.

Statistically significant differences were identified in the evaluation of ethanolic extracts against the microorganisms under evaluation (*p* < 0.0001). The findings yielded by the coagulase-positive *Staphylococcus* assay indicated that the ethanolic extracts of *R. officinalis* L. and *P. guajava* L. exhibited comparable behavior within the activity range of 21–30 mm, corresponding to Descriptor C. Among the evaluated samples, these ethanolic extracts demonstrated the most pronounced biological activity, exhibiting a tendency to the result obtained for erythromycin (D = >30 mm). Furthermore, it was determined that *C. officinalis* L., categorized as B with an activity range of 10–20 mm, exhibited a reduced level of activity in comparison to the ethanolic extracts of *R. officinalis* L. and *P. guajava* L. For the ethanolic extracts of *M. chamomilla* L. and *C. papaya* L., no zone of inhibition was observed (ZI = 0 ± 0.0). Their descriptor was A = 0–9 mm (Table 3).

In the biological evaluation of *Streptococcus* spp. (*p* < 0.0001), through the comparison of means, the ethanolic extract of *C. officinalis* L. (Descriptor C = 21–30 mm) was the extract with the best activity, approaching the ZI values presented by erythromycin used as a positive control (Descriptor D = >30 mm). In addition, it was observed that the behavior of *P. guajava* L., *R. officinalis* L., and *C. papaya* L., whose descriptor was B (ZI = 10–20 mm), presented important activity. The ethanolic extract of *M. chamomilla* L. with ZI = 0 ± 0.0 did not present activity (descriptor A = 0–9 mm), as shown in Table 3.

### 3.3. Brine Shrimp Lethality Test

A preliminary evaluation of toxicity was conducted using a multiple-proportion analysis, based on the mortality percentages of the ethanolic extracts of the selected plants. The concentrations used in this analysis were 10, 100, and 1000 µg/mL, as illustrated in Table 4. Following a 24 h period of contact between *A. salina* nauplii and ethanolic extracts, it was observed that the ethanolic extract of *C. papaya* L. resulted in 100% mortality across all concentrations tested (*p* = 1). A similar outcome was observed with the extract of *R. officinalis* L. at concentrations of 100 and 1000 µg/mL (*p* = 1). The ethanolic extract of *P. guajava* L. exhibited the lowest toxicity at all concentrations (*p* = 0.7–0.9), as did the ethanolic extract of *R. officinalis* L. at 10 µg/mL (*p* = 0.6). The findings indicate that the extracts of *P. guajava* and *R. officinalis* result in the lowest mortality rate of *A. salina* nauplii at 24 h.

## 4. Discussion

At present, diseases of microbial origin are among the most important problems facing humanity, due to the increase in the misuse of antibiotics, which generates bacterial resistance in humans and animals [44]. For this reason, it is necessary to find new sources of active pharmaceutical ingredients from plants with antibacterial pharmacological antecedents that are active against strains that cause bovine mastitis [45,46,47].

### 4.1. Phytochemical Characterization

In the preliminary phytochemical study, TLC was utilized as an analytical technique, enabling the detection of orange spots when observed under ultraviolet light (365 nm) following the application of the NP-PEG developer. This observation can be associated with the potential presence of flavonoids such as flavones, flavonols, and flavanones [31,33]. For *C. officinalis* L., it has been shown that the presence of orange spots may be associated with glycosylated flavonols such as quercetin (R_f_ = 0.4) and kaempferol (R_f_ = 0.6) [18,48]. In *P. guajava* L., two spots were observed, which were found to be related to flavonols such as myricetin (R_f_ = 0.8) and quercetin aglycone (R_f_ = 0.9) [20,26,49]. In the polar extracts of *M. chamomilla* L., flavonoids have been reported. Based on polarity and displacement on the chromatographic plate, these could be apigenin aglycone (R_f_ = 0.7) or glycosylated quercetin (R_f_ = 0.5) [50,51]. In the ethanolic extract of *C. papaya* L., the two spots could be attributed to flavonoids such as kaempferol heteroside (R_f_ = 0.3) and quercetin (R_f_ = 0.5) [24,52]. For the *R. officinalis* L. extract, two spots were identified that may correspond to flavonoids such as genkwanin (R_f_ = 0.8) and kaempferol aglycone (R_f_ = 0.7) [53,54].

### 4.2. Antibacterial Activity In Vitro of Ethanolic Extracts

Preliminary in vitro screening revealed that the ethanolic extract of *P. guajava* L. showed the best activity against a strain of coagulase-positive *Staphylococcus* (21 ± 3.2 mm). This result is comparable to that reported for the ethanolic extract of *P. guajava* L. leaves against *Streptococcus pyogenes*, with an inhibition halo of 20 mm [21]. In a separate study, the aqueous extract of *P. guajava* L. leaves was examined for its effectiveness against *S. aureus* ATCC 25923, a strain for which a smaller inhibition zone (17.67 ± 0.58 mm) was reported [55]. In a separate investigation, the antibacterial activity of the ethanolic extract of *P. guajava* L. leaves against *S. aureus* strains at a concentration of 200 mg/mL was reported [56]. In a similar vein, Eze et al. [57] reported that the ethanolic extract of *P. guajava* L. exhibited in vitro inhibitory activity against methicillin-resistant *S. aureus* (MRSA) and vancomycin-resistant *S. aureus* (VRSA). It is hypothesized that the activity observed in *P. guajava* L. may be associated with the presence of phenolic carboxylic acids, such as syringic acid and gallic acid, and/or flavonoids, including rutin, quercetin, kaempferol, and catechins, which have been reported to exhibit antibacterial activity. Furthermore, secondary metabolites were detected by TLC in this research (Figure 1) [55].

The second most effective extract against coagulase-positive *Staphylococcus* was the ethanolic extract of *R. officinalis* L. leaves (19 ± 2.1 mm), a result comparable to that reported for this same type of extract against methicillin-resistant *S. aureus* (AM 130), whose inhibition zone was 19 mm [58]. The inhibition zone obtained in this study was greater than the result obtained for the ethanolic extract of *Rosmarinus eriocalyx* against the *S. aureus* ATCC 25923 strain, whose value was 7.3 ± 0.6 [23]. Furthermore, the inhibition zone of the ethanolic extract of *R. officinalis* L. (16 mm) reported in the work of Walid et al. [59] against *S. aureus* was found to be smaller than that established in this research. The activity may be associated with the presence of phenolic carboxylic acids, such as rosmarinic acid, carnosic acid, carnosol, chlorogenic acid, and gallic acid, which have been reported in different species of *Rosmarinus*, as well as the presence of catechin and epicatechin, secondary metabolites with antibacterial activity [23,60,61,62,63].

The ethanolic extract of *C. officinalis* L. flowers demonstrated an inhibition zone of 13 ± 2.2 against coagulase-positive *Staphylococcus*. This activity is greater than that reported for the ethanolic extract of the same species evaluated against the *S. aureus* ATCC 14053 strain, whose inhibition zone was 11.03 ± 0.6 mm [64], but lower than the antibacterial activity reported for the *S. aureus* strain [JEM18] (19 ± 2 mm). The divergent outcomes of antibacterial activity may be attributable to variations in the origin of the microorganisms [19]. In the ethanolic extract of *C. officinalis* L. flowers evaluated in this research, orange spots were detected by TLC associated with flavonoids, possibly quercetin-3-o-rutinoside, kaempferol-o-rhamnosylrutinoside, isorhamnetin-3-o-glucoside, and kaempferol-o-rhamnosylrutinoside, which may be associated with biological activity [18].

The ethanolic extract of the aerial parts of *M. chamomilla* L. exhibited no biological activity against the coagulase-positive *Staphylococcus* strain, a result that is incongruent with the findings of other studies. These studies utilized an ethanolic extract obtained from flowers against various strains of *S. aureus*, and observed inhibition zones ranging from 2.5–13 mm [27,65]. This result may be associated with the yield obtained in the extraction process (8.3%), which was lower than that reported in other studies, where yields of 12.6% were reported. This factor results in a low concentration of active molecules in the ethanolic extract. Another factor that may be associated with the absence of bactericidal activity is the extraction solvent, since the substances with reported biological activity may be phenolic carboxylic acids or phenolic compounds such as glycosidic flavonoids, which are efficiently extracted with hydroalcoholic mixtures [27,65].

The ethanolic extract of *C. papaya* L. leaves demonstrated no antibacterial activity against coagulase-positive *Staphylococcus*. This outcome is incongruent with that documented by Kousar et al. [66], who utilized the methanolic extract of *C. papaya* L. leaves, yielding an inhibition zone of 26.3 mm against *S. aureus*. The inactivity of the ethanolic extracts is possibly associated with the low concentration of active metabolites such as flavonoid glycosides, tannins, and phenolic carboxylic acids, since a more polar solvent, such as methanol or hydroalcoholic mixtures, was not used [66].

Another strain associated with bovine mastitis is *Streptococcus* spp., which was evaluated against a range of extracts. The ethanolic extract of *C. officinalis* L. demonstrated the highest activity (21 ± 1.9 mm), which is comparable to that reported for the hydroethanolic extract of *Calendula tripterocarpa* against *Streptococcus mutans* (31.0 ± 0.53 mm) and *Streptococcus pyogenes* (13.4 ± 0.13 mm) [67]. A further species that has been demonstrated to be active against *Streptococcus pyogenes* is *Calendula stellata* (MIC of 10 mg/mL). This species has been associated with the presence of compounds such as quercetin and triterpene saponins, including *calendustellatosides* A–E [68]. The ethanolic extract of *R. officinalis* L. exhibited an inhibition zone measuring 17 ± 2.9 mm, indicative of enhanced activity in comparison to the essential oil of *R. officinalis* L. (10.96 ± 0.25), as evaluated against *S. agalactiae* (ATCC 12386). This observation may be attributed to the presence of flavonoids, as reported in the present study [69]. The investigation revealed that the extract of *P. guajava* L. exhibited an inhibition zone measuring 15 ± 2.3 mm, which is analogous to the results reported by Almulaiky et al. [70], who utilized a hydromethanolic extract derived from the fruit peel in their study of *S. agalactiae* ATCC 12386, yielding an inhibition zone of 13 ± 0.8 mm. This activity is associated with the presence of phenolic compounds and flavonoids. The ethanolic extract of *C. papaya* L. exhibited an inhibition zone of 14 ± 3.2 mm, classifying it as an active sample potentially associated with a flavonoid identified in this study. However, it should be noted that another study reports high activity of a papain hydrolysate against *S. aureus*, *S. agalactiae,* and *S. dysgalactiae*. Therefore, a synergistic effect between the compounds present in *C. papaya* L. could be considered [71]. The ethanolic extract of the aerial parts of *M. chamomilla* L. did not exhibit biological activity against *Streptococcus* spp., consistent with the findings of Silva et al. [72], who observed no inhibition of bacterial growth of *Streptococcus mutans* and *Streptococcus sobrinus* using the aqueous extract of *M. chamomilla* L.

### 4.3. Brine Shrimp Lethality

The lethality bioassay with *Artemia salina* is used to preliminarily evaluate the toxicity of extracts, fractions, and isolated substances from medicinal plants. The biological evaluation of the ethanolic extracts of the five plants evaluated (*C. officinalis* L., *P. guajava* L., *R. officinalis* L., *M. chamomilla* L., and *C. papaya* L.) against *A. salina* nauplii showed that their activity is toxic at the highest concentrations (100 µg/mL and 1000 µg/mL). This behavior is associated with different aspects, such as the type of extract. In our case, the ethanol used as a solvent extracts different substances of low to high polarity, among which there may be substances that generate toxicity against *A. salina* [73]. In addition, higher concentrations of substances such as terpenic saponins, alkaloids, sesquiterpenes, or phenolic compounds can generate toxicity in the organisms evaluated. An example of these compounds is the glycosides of the flavonoids quercetin and kaempferol, which showed cytotoxicity against DLD-1 colon cancer cells at a tested concentration of 200 μg/mL [74]. To reduce toxic activity, it may be worth considering evaluating aqueous or hydroalcoholic extracts, plant organs, and even the time of collection [73].

The median lethal concentration (LC50) of selected plant species has been determined by means of the *A. salina* assay in a number of studies. In the case of ethanolic and hydroalcoholic extracts of *C. officinalis* L., LC50 values of 245 µg/mL and 268.95 µg/mL have been reported, placing the activity in the medium toxicity range [75,76]. With regard to the methanol and ethanol extracts of *P. guajava* L. leaves, LC50 values of 63.81 ± 2.6 and 195 µg/mL, respectively, were reported, thus placing them within the high-toxicity range [73,77]. The ethanol extract of *R. officinalis* L. demonstrated a toxic activity level (LC50 of 36 µg/mL) [73]. Another study reported the activity of *Matricaria frigidum*, whose LC50 was 262 µg/mL [73]. Finally, the evaluation of the extract obtained by hydrodistillation of *C. papaya* L. seeds has been reported, whose LC50 was 500 µg/mL (slightly toxic) [78]. The findings of the present study are consistent with those reported above, in which the mortality of *A. salina* nauplii was observed at concentrations above 100 µg/mL. It is suggested that further experiments be conducted at concentrations below 500 µg/mL in order to obtain the LC50 of the ethanolic extracts used in this research and to establish whether the range is medium to high toxicity (LC_50_ ≤ 500 µg/mL) [73].

The findings of this study, which employed the bioassay of *A. salina*, constitute a preliminary information of the degree of toxicity of ethanolic extracts from the species *C. officinalis* L., *P. guajava* L., *R. officinalis* L., *M. chamomilla* L., and *C. papaya* L., which requires performing tests with constant concentration intervals, so that the proportionality result will be adjusted to the model and will not present atypical values. In addition, it must be complemented with other pharmacological models associated with the evaluation of in vitro toxicity with greater sensitivity, such as cell lines.

## 5. Conclusions

The results of this study demonstrated that the ethanolic extracts of *P. guajava* L. (21 ± 3.2 mm) and *R. officinalis* L. (19 ± 2.1 mm) exhibited the most significant antibacterial activity against coagulase-positive *Staphylococcus*. Furthermore, ethanolic extracts of *C. officinalis* L. (21 ± 1.9 mm), *R. officinalis* L. (17 ± 2.9 mm), and *P. guajava* L. (15 ± 2.3 mm) exhibited the most effective activity against *Streptococcus* spp., respectively. The occurrence of this phenomenon may be associated with the presence of phenolic compounds, including flavonoids, which were detected in this research by TLC. The ethanolic extract of *M. chamomilla* L. demonstrated no biological activity against the strains employed in this study. With regard to the toxicity evaluation employing the *A. salina* model, the presence of toxic activity was evidenced for all the ethanolic extracts evaluated at the highest concentrations of 100 µg/mL and 1000 µg/mL.

The findings of this research indicate the potential for the formulation of novel research proposals that build upon the bioguided fractionation of the most active ethanolic extracts. Furthermore, the evaluation of purified fractions and isolated compounds is to be conducted using the broth microdilution technique (MIC/MBC), with the objective of determining the minimum inhibitory concentration (MIC). This will facilitate the acquisition of data that validates the antibacterial activity against the microorganisms that cause bovine mastitis. It is imperative to complement this with toxicity studies of ethanolic extracts and primary fractions using in vitro models of cell lines that generate safety information, as is required in the standardization of botanical extracts. It is recommended that the chemical characterization of phenolic compounds, with a particular focus on flavonoids, be continued.

## Figures and Tables

**Figure 1 vetsci-12-00903-f001:**
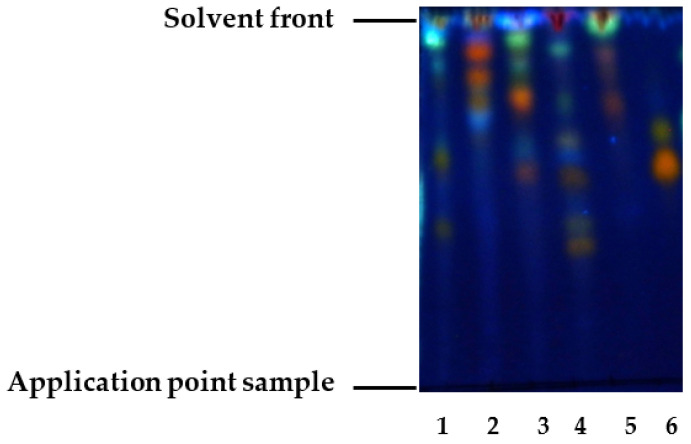
TLC chromatographic profile of ethanolic extracts of *C. officinalis* L. (1), *P. guajava* L. (2), *M. chamomilla* L. (3), *C. papaya* L. (4), *R. officinalis* L. (5), using NP/PEG as developer. Standard: (6) rutin. UV 365 nm observation; mobile phase: ethyl acetate–acetic acid-formic acid–water in proportions (100:8:4:8).

**Table 1 vetsci-12-00903-t001:** Information on plants with a history of antimicrobial use in Tolima, Colombia.

Plant Species Name	Vaucher Number	Location (Coordinates)	Collection Place	Órgano Colectado
*Calendula officinalis* L.	TOL:008702	N 4°31′9.6″–W 75°17′43.1″	Ibagué-Pastales-Vereda el “Retiro”	Flowers
*Psidium guajava* L.	TOL:009196	N 4°25′49.1″–W 75°13′2.3″	Ibagué-Universidad del Tolima	Leaves
*Matricaria chamomilla* L.	TOL:008721	N 4°31′9.6″–W 75°17′43.1″	Ibagué-Pastales-Vereda el “Retiro”	Aerial parts
*Rosmarinus officinalis* L.	TOL:005229	N 4°25′49.1″–W 75°13′2.3″	Ibagué-Pastales-Vereda el “Retiro”	Leaves
*Carica papaya* L.	TOL:005982	N 4°25′49.1″–W 75°13′2.3″	Ibagué-Universidad del Tolima	Leaves

**Table 2 vetsci-12-00903-t002:** Phytochemical characterization by TLC for the detection of flavonoids in ethanolic extracts.

Ethanolic Plant Extract	R_f_ ^a^ Values	^b^ Flavonoids (Green or Orange)
*C. officinalis* L.	0.6, 0.8	Orange
*P. guajava* L.	0.8, 0.9	
*M. chamomilla* L.	0.4, 0.5, 0.6	
*R. officinalis* L.	0.4, 0.6	
*C. papaya* L.	0.7, 0.8	

R_f_ ^a^ values obtained from chromatographic tests by TLC, ^b^ fluorescence at 365 nm. Standard: rutin (0.6 orange).

**Table 3 vetsci-12-00903-t003:** Antibacterial activity of ethanolic extracts against coagulase-positive *Staphylococcus* and *Streptococcus* spp. after 24 h.

Bacterial Strain	Antibacterial Activity of Ethanolic Extracts of Selected Plants by Evaluating the Zone of Inhibition (ZI) in mm * ± Standard Deviation						
	*C. officinalis* L.	*P. guajava* L.	*M. chamomilla* L.	*R. officinalis* L.	*C. papaya* L.	Positive control erythromycin	Negative control DMSO 10%
Coagulase-positive *Staphylococcus*	13 ± 2.2 ^B^	21 ± 3.2 ^C^	0 ± 0.0 ^A^	19 ± 2.1 ^C^	0 ± 0.0 ^A^	32 ± 4.5 ^D^	0 ± 0.0
*Streptococcus* spp.	21 ± 1.9 ^C^	15 ± 2.3 ^B^	0 ± 0.0 ^A^	17 ± 2.9 ^B^	14 ± 3.2 ^B^	31 ± 4.5 ^D^	0 ± 0.0

* Mean of three independent experiments, each with five repetitions. Data marked with capital letters as superscripts (A–D) correspond to the comparison of means. Means with a common letter are not significantly different (*p* > 0.05).

**Table 4 vetsci-12-00903-t004:** Mortality ratio by concentration of each ethanolic extract of the plant species against nauplii of *A. salina* after 24 h.

Concentration (µg/mL)	*P. guajava* L.	*R. officinalis* L.	*C. papaya* L.	*C. officinalis* L.	*M. chamomilla* L.
1000	0.73	1.0	1.0	0.96	0.96
100	0.80	1.0	1.0	0.96	0.96
10	0.90	0.60	1.0	1.0	0.76

## Data Availability

The data that were generated and analyzed in this study are presented in their entirety in this article, thus facilitating the verification and validation of the findings.

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
