# Peer review of "In Vitro Antibacterial Activity of Ethanolic Extracts Obtained from Plants Grown in Tolima, Colombia, Against Bacteria Associated with Bovine Mastitis"

_vetsci, 2025, doi:10.3390/vetsci12090903_

Round 1

Reviewer 1 Report

Comments and Suggestions for Authors

In Vitro Antibacterial Activity of Ethanolic Extracts from Plants Grown in Tolima, Colombia, Against Bacteria Associated with Bovine Mastitis

Line #

Comments

The manuscript addresses critical need for alternatives to antibiotics in bovine mastitis, however it requires significant modifications as per following suggestions:

Methods

Can you specify the positive and negative controls during AST experiments?

Is it same for all pathogens?

Introduction

It must be improved keeping in view the importance of mastitis, pathogens, AMR, and alternative therapeutic agents.

The introduction part is much shorter and with only 9 references from previous studies. It must be supported by sufficient recent references.

Some of the following may help to emphasize the importance of mastitis and its causative agents.

Complete Genome of Multi-Drug Resistant Staphylococcus aureus in Bovine Mastitic Milk in Anhui, China
Junjun Liu, Xinglin Zhang, Jianrui Niu, Zhaoqing Han, Chongliang Bi, Khalid Mehmood, Dunia A Al Farraj, Inshad Alzaidi, Rashid Iqbal and Jianhua Qin
Pak Vet J, 2023, 43(3): 456-462
Abstract  Full text:   pdf

Epidemiological Features, Biochemical Indices, Antibiogram Susceptibility Profile and Biofilm Factor Genes of Klebsiella pneumoniae Isolated from Bovine Clinical Mastitis Cases
Ayman Ahmed Shehata, Eman Beshry Abd-Elfatah, Hend E.M. Elsheik, Asmaa lbrahim Abdelaziz Zin Eldin, Marwa B Salman, Ahmed Shehta, Safaa I Khater and Mahran Mohamed Abd El-Emam
Pak Vet J, 2024, 44(1): 141-147
Abstract  Full text:   pdf

Prevalence of Subclinical Mastitis in Holstein-Friesian Cow Dairy Among Small-Scale Farms in Batu, Indonesia

Rifa’i, Lilik Eka Radiati, La Choviya Hawa and Puguh Surjowardojo

Int J Vet Sci, 2024, 13(5): 592-595.

Determination of Subclinical Mastitis Prevalence in Dairy Cows in Türkiye through Meta-Analysis and Production Loss Calculation
Mehmet Saltuk Arikan, Burak Mat, Hasan Alkan, Mustafa Bahadır Çevrimli, Ahmet Cumhur Akin, Ebru Kaya BaÅŸar and Mustafa Agah Tekindal
Pak Vet J, 2024, 44(2): 391-399
Abstract  Full text:   pdf

Negar GHAZVINEH, Azam MOKHTARI, et. al., Molecular Detection of Selective Virulence Factors of Mycoplasma bovis Local Isolates Involved in Bovine Mastitis. Kafkas Univ Vet Fak Derg, 30 (5): 631-639, 2024.

https://doi.org/10.9775/kvfd.2024.32118

Methods

Specify bacterial identification methods

Perform genetic identification and antibiotic susceptibility testing on isolates. Report resistance profiles of the isolated strains.

Approval from Biosafety/ Bioethics committee?

Is the Kirby-Bauer disc diffusion method  validated for plant extracts (CLSI guidelines)?

Can you repeat assays using broth microdilution (MIC/MBC) and include controls for solvent effects (DMSO). Clarify if CLSI/EUCAST standards were followed.

Can you specify McFarland concentration of bacterial suspension used?

Results of antimicrobial activity revealed in mm with SD,

Can you specify the number of replicates for each experiment.

Can we include results (pictures) of antimicrobial activity of various extracts?

Fig. 1

Can we have a clearer image?

Table 3

DMSO is positive or negative control?

 Antimicrobial Activity or antibacterial activity?

Both are used to describe the same results.

Why can’t you include MRSA strains from isolated strains?

Can you include in vitro cytotoxicity (e.g., bovine mammary epithelial cells) and clarify limitations of A. salina assay.

TLC detection of flavonoids is qualitative and non-specific. No quantification (e.g., HPLC) or bioassay-guided fractionation links flavonoids to bioactivity.

Does Artemia salina lethality (24-hr exposure) a good predictor of mammalian toxicity?

Can you add HPLC quantification of major flavonoids (e.g., quercetin, kaempferol) and correlate levels with bioactivity?

Author Response

2. Point-by-point response to Comments and Suggestions for Authors

Introduction

Comments 1: It must be improved keeping in view the importance of mastitis, pathogens, AMR, and alternative therapeutic agents

Response 1: Observation accepted. The introduction included the following aspects: the importance of mastitis, the pathogens, antimicrobial resistance (AMR), and alternative therapeutic agents (Lines: 55-120).

Comments 2: The introduction part is much shorter and with only 9 references from previous studies. It must be supported by sufficient recent references. Some of the following may help to emphasize the importance of mastitis and its causative agents.

Response 2: Observation accepted. The references recommended by the reviewer were used, and the introduction was expanded (Lines 55-142).

Methods

Comments 3: Specify bacterial identification methods

Response 3: Observation accepted. Microbiological Culture: The samples were inoculated using the surface streaking method with a calibrated loop (0.01 mL) onto blood agar and MacConkey agar media. They were incubated for 24 to 48 hours at 37°C under aerobic conditions. The isolates were subjected to colony morphology analysis and Gram staining for morphological evaluation. Catalase testing was performed on Gram-positive cocci to differentiate Streptococcus from Staphylococcus. The tube coagulase test was employed to differentiate S. aureus and other coagulase-positive staphylococci (CPS) from coagulase-negative staphylococci (CNS). Gram-positive cocci, most of which exhibited dew-drop-like growth on blood agar and were catalase-negative, were identified as Streptococcus spp. For Gram-negative colonies, the following biochemical tests were performed: indole, methyl red, Voges-Proskauer, Simmons citrate, and triple sugar iron (TSI) agar. Subsequently, antibiotic susceptibility testing was conducted on CPS isolates using trimethoprim-sulfamethoxazole (25 μg), penicillin G (10 IU), cefoperazone (75 μg), streptomycin (10 μg), tetracycline (30 μg), and erythromycin (15 μg). Cultures were classified as susceptible (S), intermediate (I), or resistant (R) based on the diameter of the inhibition zone. In addition, novobiocin (5 μg) was used for classification as novobiocin-susceptible or novobiocin-resistant. These tests were performed using the Mueller-Hinton agar diffusion method, following the guidelines established (Lines 203-221).

Comments 4: Perform genetic identification and antibiotic susceptibility testing on isolates. Report resistance profiles of the isolated strains.

Response 4: Observation accepted. Genetic identification was not performed as the isolates were phenotypically grouped as coagulase-positive Staphylococcus.

Gram-positive cocci exhibiting dew-drop-like growth on blood agar and catalase-positive reactions were associated with Streptococcus spp.

Gram-positive cocci arranged in clusters, isolated from colonies that grew on blood agar but not on MacConkey agar, showing a golden color, beta-hemolysis, and catalase-positive reactions, were associated with Staphylococcus. The coagulase-positive isolates were tested against the extracts (Lines 223-230).

Comments 5: Approval from Biosafety/ Bioethics committee?  

Response 5: Observation accepted. Sample collection was carried out during the morning milking, and no experimental procedures were performed; for this reason, approval from an ethics committee was not required. The study was conducted with informed consent for sample collection (Lines: 232-234). (Informed consent document attached).

Comments 6: Is the Kirby-Bauer disc diffusion method validated for plant extracts (CLSI guidelines)?

Response 6: Observation accepted. The use of plant extracts is an alternative method for the treatment of infectious diseases, specifically in cases such as bovine mastitis. Although there is currently no standardized plant extract for this purpose, the Kirby-Bauer technique, approved by the CLSI, is employed using sensidiscs impregnated with the antibiotic or substance to be evaluated, allowing for the determination of its activity against bacteria (Lines: 239-256).

Comments 7: Can you repeat assays using broth microdilution (MIC/MBC) and include controls for solvent effects (DMSO). Clarify if CLSI/EUCAST standards were followed.

Response 7: Observation accepted. The broth microdilution technique (MIC/MBC) was not performed in this study, as it was not included within the specific objectives of the project. However, in future research, the inclusion of this methodology could be considered in order to determine the minimum inhibitory concentration (MIC) of the compounds and extracts under study. (Lines 569-579)

Comments 8: Can you specify McFarland concentration of bacterial suspension used?

Response 8: Observation accepted. A 0.5 McFarland standard was used, which corresponds to 1.5 x 10^8 (150 million) microorganisms per milliliter, equivalent to 1.5 x 10^8 CFU/mL. (Lines 244-246).

Comments 9: Results of antimicrobial activity revealed in mm with SD, Can you specify the number of replicates for each experiment.

Response 9: Observation accepted. Three independent experiments were performed, each with five (5) repetitions for each bacterial strain (Lines 261-269).

Comments 10: Can we include results (pictures) of antimicrobial activity of various extracts?

Response 10: Observation accepted. The images showing the antimicrobial activity of the five extracts were included in the supplementary material: “Figure S1. Inhibition zones of ethanolic extracts: M. chamomilla, C. officinalis, R. officinalis, P. guajava and C. papaya. Positive control (C+) (Erythromycin 10 mg/mL). Coagulase positive Staphylococcus (A) and Streptococcus spp (B).”

Comments 11: Can we have a clearer image?

Response 11: Observation accepted. The image quality has been improved (Line: 318).

Comments 12: DMSO is positive or negative control?

Response 12: Observation accepted. DMSO is the negative control (Table 3) (Line: 395).

Comments 13: Antimicrobial Activity or antibacterial activity? Both are used to describe the same results.

Response 13: Observation accepted. The term "antibacterial activity" was used throughout the document, as it specifically refers to the inhibition of bacterial growth.

Comments 14: Why can’t you include MRSA strains from isolated strains?

Response 14: Observation accepted. MRSA strains were not included because methicillin is not used intramammary for the treatment of bovine mastitis in Colombia.

Comments 15: Can you include in vitro cytotoxicity (e.g., bovine mammary epithelial cells) and clarify limitations of A. salina assay.

Response 15: For the scope of this research, in vitro cytotoxic evaluation using bovine mammary epithelial cells was not considered, as this requires method standardization. Instead, a preliminary toxicity assessment was proposed using a simple and cost-effective model within our reach, such as the Artemia salina lethality bioassay. Preliminary safety data on the primary extracts of medicinal plants were sought.

The limitations of the A. salina assay were considered in the analysis of the results.

Comments 16: TLC detection of flavonoids is qualitative and non-specific. No quantification (e.g., HPLC) or bioassay-guided fractionation links flavonoids to bioactivity.

Response 16: Observation accepted. As an initial stage, the detection of flavonoids was performed under conditions reported in the literature and routinely used in our laboratory (mobile phase: ethyl acetate - acetic acid - formic acid - water in proportions of 100:8:4:8). Rutin was used as the reference substance. The detection reagents were diphenylboric acid 2-aminoethyl ester (NP reagent) and polyethylene glycol 4000 (PEG 4000). These conditions are widely used for the preliminary characterization of flavonoids, confirming color patterns and migration behavior on the plate. The quantification of flavonoids was not proposed in this study. Based on the results obtained in this research, it is recommended to continue with bio-guided fractionation of the extracts showing the best biological activity and to further explore chemical studies focused on phenolic compounds, particularly flavonoids. (Lines 569-579)

Comments 17: Does Artemia salina lethality (24-hr exposure) a good predictor of mammalian toxicity?

Response 17: Observation accepted. The Artemia salina model is a preliminary bioassay and is not specific for antitumor activity or any particular physiological action. However, it is important as it provides preliminary safety information on primary extracts from medicinal plants, which should later be confirmed using more robust and standardized models, such as different cell lines. The evaluation of toxicity in cell lines was not an objective of this study; such assessments will be conducted with the extracts showing the best antibacterial activity (Lines 548-554).

Comments 18: Can you add HPLC quantification of major flavonoids (e.g., quercetin, kaempferol) and correlate levels with bioactivity?

Response 18: Quantification of flavonoids by HPLC was not performed. In our research, we conduct the quantification of secondary metabolites, such as flavonoids, during the standardization of extracts that have previously demonstrated the best biological activity and have undergone safety studies. (Lines 569-579).

Reviewer 2 Report

Comments and Suggestions for Authors

The present article, entitled "In vitro antibacterial activity of ethanolic extracts from plants grown in Tolima, Colombia, against bacteria associated with bovine mastitis", aims to evaluate potential antimicrobial effect of natural compounds extracted from plants cultivated in Colombia.

Even though the research topic is of utmost importance, given the non stop developement of antibiotic resistance in microbial pathogens, the article itself lacks some of the scientific rigor necessary to present this kind of data and needs extensive improvements both in study design and drafting of the article.

Below are listed, in a line by line format, the suggestions/required modifications for the authors.

Line 6: please remove the URL of the orcid id

Line 27-52: all the tested plants of this study should be mentioned in the abstract

Line 47-52: repetition of what is stated in lines 39-43, please remove these lines.

Line 53-83: repetition of the entire sections abstract and keywords, please remove.

Line 98-104: this paragraph should be included in the results section, not the introduction since those are results that the authors obtained and are not retrieved from literature. Also these results need to be contextualized in this article, methods used need to be adequately described and the breakpoints used to determine resistance need to be explicitely written since there is not a single international standard for antimicrobial resistance testing and evaluation.

Line 88-116: Introduction is really short, authors should consider to exapand it with more informations retrieved from already published literature.

Line 120: "previous reports on antimicrobial activity" please explicitly cite those reports.

Section 2.4.3: Statistical analysis section needs to be extensively revised.

  1. were the variables tested for normal distribution? if yes with what test and why that test was chosen?
  2. the three independent experiments were carried out with different isolates of S. aureus and S. agalactiae? why? In the opinion of the reviewer trying to compare the efficacy of the natural compounds while introducing variability of the isolates is wrong because the effect seen needs to be attributed not only to the efficacy of the compound but also to the different fittnes of the different isolates used.
  3. what is the achieved power of the test on the primary outcome

Results section: this section is lacking the results of the isolation of S. aureus and S. agalactiae colonies from the 100 milk samples mentioned at line 149. Moreover no results are presented on antimicrobial resistance testing mentioned in introduction (line 98-104). These results need to be included in this section. No information about the characterization of antimicrobial resistance of the isolates used to test the efficacy of the plant extracts is provided, but it should be included.

Section 2.5: even if the authors cite a previous paper a brief description of the methodology should be included in the text.

Table 3: "positive control DMSO 10%" should be "negative control DMSO 10%" instead.

Section 3.3: mortality ratios of P. guajava L. seem odd, logical thinking would lead to believe that an higher concentration of the molecule would be more toxic, here the trend is inversed and the authors should comment on it in the discussion section (4.3).

Comments on the Quality of English Language

The overall quality of the english is poor and needs complete revision by the authors in grammar, synthax, spelling and overall fluency of the text.

The reviewer advises the revision of the text by a native english speaker or by english editing services.

Author Response

3. Point-by-point response to Comments and Suggestions for Authors

Comments 1: Line 6: please remove the URL of the orcid id

Response 1: Observation accepted. ORCID was removed (line 6).

Comments 2: Line 27-52: all the tested plants of this study should be mentioned in the abstract.

Response 2: Observation accepted. The scientific names of the evaluated plants have been included in the abstract (Lines 34-36).

Comments 3: Line 47-52: repetition of what is stated in lines 39-43, please remove these lines.

Response 3: Observation accepted. The suggested lines have been removed.

Comments 4: Line 53-83: repetition of the entire sections abstract and keywords, please remove.

Response 4: Observation accepted. The repeated abstract and keywords sections have been removed.

Comments 5: Line 98-104: this paragraph should be included in the results section, not the introduction since those are results that the authors obtained and are not retrieved from literature. Also these results need to be contextualized in this article, methods used need to be adequately described and the breakpoints used to determine resistance need to be explicitely written since there is not a single international standard for antimicrobial resistance testing and evaluation.

Response 5: Observation accepted. The information cited by the reviewer corresponds to the data reported in the references mentioned in the text (lines: 98, 105).

Comments 6: Line 88-116: Introduction is really short, authors should consider to exapand it with more informations retrieved from already published literature.

Response 6: Observation accepted. The introduction has been expanded to include aspects such as the importance of mastitis, pathogens, antimicrobial resistance (AMR), and alternative therapeutic agents (55-91).

Comments 7: Line 120: "previous reports on antimicrobial activity" please explicitly cite those reports.

Response 7: Observation accepted. References related to antibacterial activity have been included (Lines 146-147).

Comments 8: Section 2.4.3 Statistical analysis section needs to be extensively revised.

Response 8: Observation accepted. Normality tests were performed and evaluated to assess the distribution of the data. Subsequently, an analysis of variance (ANOVA) was conducted to identify statistically significant differences between the treatments. Tukey’s test was then employed to perform a comparative evaluation among the extracts and to determine which extracts exhibited similar or different activities (means sharing the same letter are not significantly different, p > 0.05) (Lines 261-269).

Example:

Comments 9: the three independent experiments were carried out with different isolates of S. aureus and S. agalactiae? why? In the opinion of the reviewer trying to compare the efficacy of the natural compounds while introducing variability of the isolates is wrong because the effect seen needs to be attributed not only to the efficacy of the compound but also to the different fittnes of the different isolates used.

Response 9: Observation accepted. Three independent experiments were conducted for S. aureus and S. agalactiae strains, which were obtained from the same matrix, not from different matrices.

Comments 10: What is the achieved power of the test on the primary outcome

Response 10: Observation accepted. If the term “power” is associated with the greatest activity (efficiency), the initial screening determined that the extracts of R. officinalis L. and P. guajava L. exhibited the highest activity against S. aureus, while the ethanolic extract of C. officinalis was the most active against S. agalactiae.

Comments 11: Results section: this section is lacking the results of the isolation of S. aureus and S. agalactiae colonies from the 100 milk samples mentioned at line 149. Moreover no results are presented on antimicrobial resistance testing mentioned in introduction (line 98-104). These results need to be included in this section. No information about the characterization of antimicrobial resistance of the isolates used to test the efficacy of the plant extracts is provided, but it should be included.

Response 11: The results corresponding to the isolation of Staphylococcus aureus and Streptococcus agalactiae colonies from 100 cows come from a previous study published under the reference 34 “Prevalence of bovine mastitis in the Anaime Canyon, a Colombian dairy region, including etiology and antimicrobial resistance”, and are not part of the original results presented in the present work (Line: 194).

Comments 12: Section 2.5: even if the authors cite a previous paper a brief description of the methodology should be included in the text.

Response 12: Observation accepted. The description of the Brine Shrimp Lethality Bioassay has been provided (Lines 281-292).

Comments 13: Table 3: "positive control DMSO 10%" should be "negative control DMSO 10%" instead.

Response 13: Observation accepted. It has been corrected in (Table 3) (Lines 375).

Comments 14: Section 3.3:  mortality ratios of P. guajava L. seem odd, logical thinking would lead to believe that an higher concentration of the molecule would be more toxic, here the trend is inversed and the authors should comment on it in the discussion section (4.3).

Response 14: Observation accepted. The discussion regarding the mortality rate has been expanded, and it is proposed that future studies use smaller concentration intervals, as wide intervals may lead to this type of behavior with mortality affecting only a single individual.

3. Point-by-point response to Comments and Suggestions for Authors

Reviewer 3 Report

Comments and Suggestions for Authors

Some phrases and grammatical structures are awkward. A thorough language revision by a native English speaker or professional editing service is recommended.

All visual elements are clear and support the findings well. Consider providing higher-resolution versions of TLC images for future publication.

While antimicrobial activity is demonstrated, the high toxicity reported in Artemia salina is significant. The authors should more explicitly address this limitation and possible strategies to mitigate it in the conclusions.

The paper would benefit from a more detailed discussion about how these extracts can be further developed into safe therapeutic agents. 

Please strengthen the scientific background and contextual support for the use of natural products as alternatives to antibiotics in the treatment of bovine mastitis.

This study reports relevant findings on the in vitro antibacterial properties of propolis against mastitis-related pathogens and would contribute to enhancing the depth and relevance of your discussion.

Line 106 – In the Introduction, after the mention of plant-derived antimicrobial alternatives.

Line 291 – In the Discussion, Section 4.2, after discussing the activity of Psidium guajava L. and other extracts.

Incorporating this reference would improve the contextual quality of the manuscript by highlighting additional evidence supporting natural antibacterial agents relevant to your study. Thank you for considering this addition.

The references to supplementary tables/figures are appropriate and provide necessary context.

Comments on the Quality of English Language

The English language is generally understandable but contains grammatical and structural inconsistencies that could be improved to enhance clarity and scientific precision.

Author Response

  1. Point-by-point response to Comments and Suggestions for Authors:

Comments 1: Some phrases and grammatical structures are awkward. A thorough language revision by a native English speaker or professional editing service is recommended.

Response 1: Se accept la observación. The final text was reviewed for style at our language center by an English language professional.

Comments 2: All visual elements are clear and support the findings well. Consider providing higher-resolution versions of TLC images for future publication.

Response 2: Observation accepted. The figure is presented at 900 DPI, as suggested by the journal (Line 318).

Comments 3: While antimicrobial activity is demonstrated, the high toxicity reported in Artemia salina is significant. The authors should more explicitly address this limitation and possible strategies to mitigate it in the conclusions.

Response 3: Observation accepted. In the conclusions, it is recommended to evaluate the toxicity of ethanolic extracts and primary fractions using in vitro cell line models in order to generate safety data (Lines 565-567).

Comments 4: The paper would benefit from a more detailed discussion about how these extracts can be further developed into safe therapeutic agents. 

Response 4: Observation accepted. The discussion was extended to the efficacy and safety aspects of the medicinal plants used in this research. In addition, chemical and pharmacological studies oriented to standardized extracts were suggested. (Lines:424-508; 575-579)

Comments 5: Please strengthen the scientific background and contextual support for the use of natural products as alternatives to antibiotics in the treatment of bovine mastitis.

Response 5: Observation accepted. In the introduction section, the advantages of natural products as alternatives to antibiotics are mentioned, as they reduce the duration of treatment, act synergistically through different mechanisms of action, and are less toxic (Lines 109-120).

Comments 6: This study reports relevant findings on the in vitro antibacterial properties of propolis against mastitis-related pathogens and would contribute to enhancing the depth and relevance of your discussion.

Response 6: In this study, the ethanolic extracts of Calendula officinalis L., Psidium guajava L., Matricaria chamomilla L., Rosmarinus officinalis L., and Carica papaya L. were evaluated (Lines 144-256).

Comments 7: Line 106 – In the Introduction, after the mention of plant-derived antimicrobial alternatives.

Response 7: Observation accepted regarding the introduction. It has been adjusted in the document.

Comments 8: Line 291 – In the Discussion, Section 4.2, after discussing the activity of Psidium guajava L. and other extracts.

Response 8: Observation accepted. It has been adjusted in the document.

Comments 9: Incorporating this reference would improve the contextual quality of the manuscript by highlighting additional evidence supporting natural antibacterial agents relevant to your study. Thank you for considering this addition.

Response 9: Observations accepted, and updated references have been attached.

Comments 10: The references to supplementary tables/figures are appropriate and provide necessary context.

Response 10: Observations accepted. Supplementary material has been attached along with the article.

Comments 11: Unclear phrasing and repetitive statements. Clarify the link between the studied plants and the targeted bacteria.

Response 11: Observations accepted. It was mentioned in the document that these plants have ethnopharmacological information related to the treatment of bacterial diseases; this was the criterion for selecting the species in this study (Lines: 139-142)

Comments 12: Redundancy between the “Simple Summary” and the Abstract; some vague or repetitive expressions. Streamline both sections to avoid duplication. Focus on scientific clarity and concise language

Response 12: Observations accepted. The corrections have been made in the Simple Summary.

Comments 13: Overly long sentences and weak logical transitions. Break into shorter sentences and clearly connect causes and effects.

Response 13: Observations accepted. The corrections have been made in the introduction. (Lines 55-142)

Comments 14: List of resistance rates is difficult to follow and lacks summarizing conclusion. Reformat the list for readability and add a final sentence summarizing the impact of resistance.

Response 14: Observations accepted. The corrections have been made in the introduction (Lines 55-142)

Comments 15: Vague wording like “have demonstrated bactericidal properties” without precise references. Link this claim more directly to specific references and describe under what conditions those results were observed.

Response 15: Observations accepted. The plants, pharmacologically relevant metabolites, and the biological activity associated with their corresponding references were described. (Lines 122-133)

Comments 16: Ambiguous language in the extraction method description. Clarify each extraction step. Replace phrases like “exhaustion of the material was confirmed by TLC” with more precise scientific wording.

Response 16: Observations accepted. The adjustments requested by the reviewer have been made (156-174)

Comments 17: TLC methodology is difficult to follow due to awkward phrasing. Please use standard scientific terminology and improve clarity, e.g., “Flavonoids were detected under UV light at 365 nm after development with NP-PEG reagent.”

Response 17: Observations accepted. The adjustments requested by the reviewer have been made. (Lines: 180-188)

Comments 18: - Instructions are overly detailed and contain informal terms (e.g., “mass sweeping”). Please use standard microbiology vocabulary and simplify the procedural explanation.

Response 18: Observations accepted. The term 'mass sweeping' was replaced by 'sweeping technique' in the article, as the latter is more in line with the technical and standardised vocabulary used in the field of microbiology. (Lines: 190-256)

Comments 19: Repetition of data already stated in the Abstract. Please focus more on interpretation of the results and less on repeating raw numbers.

Response 19: Observations accepted. The relevant results of the research are reported in the abstract. In the Results section, the findings are presented in greater detail along with their interpretation. Subsequently, in Section 4, the results are discussed (Lines: 335-353)

Comments 20: Confusing correlation between activity levels (A–D descriptors) and extract efficacy. Please clarify the classification system and ensure consistency in how it relates to antimicrobial performance. (Lines: 355-373)

Response 20: Observation accepted. Corrections have been made to the writing.

Comments 21: Repetition of results without deeper interpretation. Add insight into why certain flavonoids may explain the observed antibacterial activity.

Response 21: In section 4.1 Phytochemical Characterization, the TLC results are correlated with the presence of flavonoids as secondary metabolites of interest in each sample. In section 4.2 In Vitro Antibacterial Activity of Ethanolic Extracts, the biological activity is related to the secondary metabolites of interest, including flavonoids (Lines: 424-508)

Comments 22: - Inconsistent connection between Rf values and their associated flavonoids. Clearly match each spot with its corresponding flavonoid, supported by literature.

Response 22: Observation accepted. The discussion has been expanded by incorporating the structural characteristics of the molecules, their migration on the plate as indicated by the orange spots, and the references were correlated with the Rf values. (Lines: 407-420)

Comments 23: Weak justification for lack of antimicrobial activity; vague explanation about solvent effects. Discuss polarity, extract concentration, and how these influence compound solubility more clearly.

Response 23: Observation accepted. The discussion has been expanded to incorporate aspects related to the concentration of active molecules and polarity (Lines: 424-508)

Comments 24: Ambiguity in interpreting toxicity; LCâ‚…â‚€ threshold not explained. Define “biologically relevant toxicity” (e.g., LCâ‚…â‚€ < 1000 µg/mL) and relate this clearly to your findings.

Response 24: Observation accepted. The discussion has been expanded to incorporate interpretative aspects of toxicity according to the biological model and the results obtained (Lines: 512-554).

Comments 25: Conclusion is too long and somewhat repetitive. Split into two paragraphs: one for findings, one for future research directions. Use concise language.

Response 25: Observation accepted. The paragraphs have been separated, with one presenting the findings and the other outlining future lines of research (Lines: 557-579).

Round 2

Reviewer 1 Report

Comments and Suggestions for Authors

Line 86: staphylococcus

Staphylococcus

Ref. No. 4: incomplete, please correct it.

Ref. No. 10: Incomplete and is not English, English reference is available, please update.

Please ensure the remaining references too.

Best wishes

Author Response

Comments 1: - Line 86: staphylococcus.

Response 1: Observation accepted. Corrected in document (Line:86)

Comments 2. Ref. No. 4: incomplete, please correct.

Response 2: Observation accepted. Corrected Ref No 4 in document (Line:610)

Comments 3: Ref. No. 10: Incomplete and is not English, English reference is available, please update

Response 3: Observation accepted. Corrected Ref. No. 10 in document (Line: 627)

Comments 4: Please correct also the rest of the references.

Response 4: Observation accepted. All references have been revised.

Reviewer 2 Report

Comments and Suggestions for Authors

The authors answered to most of the comments, nevertheless two comments were not addressed.

comment 10: the reviewer is asking to explicitly state the statistical power calculation, without this information the results of a statistical test (hypotesis testing in this specific case) cannot be considered reliable.

comment 11: if results are presented in another work the their relative materials and methods part should not be included in the present

Author Response

Comments 1:  the reviewer is asking to explicitly state the statistical power calculation, without this information the results of a statistical test (hypotesis testing in this specific case) cannot be considered reliable.

Response 1: Observation is accepted. The statistical methodology used was a completely randomized design with one-way analysis of variance (ANOVA) at 95% confidence, with a significance level of α=0.05 (5%), and statistical tests were performed to determine the normality of the data. It was considered that with a value P<0.05 the null hypothesis (H0) was rejected (Lines: 255-263;351-369).

Analysis of variance of ethanolic extracts against coagulase positive Staphylococcus

Analysis of variance of ethanolic extracts against Streptococcus spp.

Comments 2. if results are presented in another work the their relative materials and methods part should not be included in the present

Response 2: Observation accepted. It is clarified in the document that the collection of biological material and the isolation of microorganisms was carried out in the research conducted by the Microbiologist Pilar Sánchez, professor at the Universidad Cooperativa de Colombia.  However, the authors of this research consider it appropriate to describe the process of collection of biological material and isolation of microorganisms used in this research, citing the primary source of information (Lineas: 193-204).

In this work, resistance tests were not performed, the paragraph Resistance tests is deleted: “Subsequently, coagulase positive Staphylococcus (CPE) isolates were subjected to an-tibiotic susceptibility testing, employing the Müller Hinton agar diffusion method in ac-cordance with the guidelines stipulated by the Clinical and Laboratory Standards In-stitute (38,39). The following antimicrobial discs were utilised: trimethoprim sulfamethoxazole (25 μg), penicillin G (10 IU), cefoperazone (75 μg), streptomycin (10 μg), tetracycline (30 μg), and erythromycin (15 μg). The outcomes were interpreted as sensitive (S), intermediate sensitivity (I) or resistant (R), based on the diameter of the inhibition zone. Furthermore, novobiocin (5 μg) was utilised for the classification of samples as novobiocin sensitive or novobiocin-resistant”.
